# Midwives speaking out on COVID-19: The international confederation of midwives global survey

Donna L. Hartz[1,2‡]*, Sally K. Tracy[2,3‡], Sally Pairman[4], Ann Yates[4], Charlotte Renard[4], Pat Brodie[5], Sue Kildea[2]

1 University of Newcastle, Gosford, Australia, 2 Molly Wardaguga Research Centre, Charles Darwin University, Alice Springs, Casuarina, Australia, 3 University of Sydney, Sydney, Australia, 4 International Confederation of Midwives (ICM), The Hague, The Netherlands, 5 University of Technology Sydney, Ultimo, Australia

‡ DLH and SKT are joint senior authors on this work.
* Donna.Hartz@newcastle.edu.au

## Abstract

### Background

Maternity services around the world have been disrupted since the outbreak of the COVID-19 pandemic. The International Confederation of Midwives (ICM) representing one hundred and forty-three professional midwifery associations across the world sought to understand the impact of the pandemic on women and midwives.

### Aim

The aim of this study was to understand the global impact of COVID-19 from the point of view of midwives' associations.

### Methods

A descriptive cross-sectional survey using an on-line questionnaire was sent via email to every midwives' association member of ICM.

### Survey instrument

The survey was developed and tested by a small global team of midwife researchers and clinicians. It consisted of 106 questions divided into seven discreet sections. Each member association was invited to make one response in either English, French or Spanish.

### Results

Data were collected between July 2020 and April 2021. All respondents fulfilling the inclusion criteria irrespective of whether they completed all questions in the survey were eligible for analysis. All data collected was anonymous. There were 101 surveys returned from the 143 member associations across the world. Many countries reported being caught unaware of the severity of the infection and in some places, midwives were forced to make their own

**Data Availability Statement:** Data are within the paper and its Supporting Information files. Within the paper there are deidentified data, both descriptive quantitative and explicit qualitative text

quotes, to support the results of this research project. The data collection survey tool has also been provided as Supporting Information files.

**Funding:** Contracted project by United National Population Fund (UNFPA): https://www.unfpa.org UNFPA played no direct role in the undertaking of this research. A UNFPA representative (Dr Bar-Zeev) was member of the steering committee and due to her unique role in maternal health in UNFPA contributed to the study design and survey tool development.

**Competing interests:** The authors have declared that no competing interests exist.

PPE, or reuse single use PPE. Disruption to maternity services meant women had to change their plans for place of birth; and in many countries maternity facilities were closed to become COVID-19 centres. Half of all respondents stated that women were afraid to give birth in hospitals during the pandemic resulting in increased demand for home birth and community midwifery. Midwifery students were denied access to practical or clinical placements and their registration as midwives has been delayed in many countries. More than 50% of the associations reported that governments did not consult them, and they have little or no say in policy at government levels. These poor outcomes were not exclusive to high-, middle- or low-income countries.

## Conclusions

Strong recommendations that stem from this research include the need to include midwifery representation on key government committees and a need to increase the support for planned out of hospital birth. Both these recommendations stand to enhance the effectiveness of midwives in a world that continues to face and may face future catastrophic pandemics.

## Background

The COVID-19 pandemic has placed the entire world's population in grave danger. On March 11, 2020, the Director of the World Health Organisation issued the following warning:

"This is not just a public health crisis; it is a crisis that will touch every sector . . . countries must take a 'whole of government,' 'whole of society' approach, built around a comprehensive strategy to prevent infections, save lives, and minimise impact" [1]. As the virus spread rapidly throughout the world it exposed inequality between and within countries. As it gained momentum, health systems faced increasing pressures and deaths increased. Countries were urged to strike a balance between protecting health, preventing economic and social disruption, and respecting human rights [2].

At the outbreak of the pandemic very little evidence was available to inform maternity care providers on the severe acute respiratory syndrome–coronavirus-2 (SARS-COV-2). Consequently, maternity systems throughout the world were disrupted forcing many countries to implement policies that restricted support for labouring mothers, leaving women feeling isolated and fearful [3, 4]. The potential concern for transmission of infection from maternal respiratory secretions to the newborn led in some instances, to the temporary separation of mothers and babies. This included prevention of breastfeeding until more evidence became available to support the benefits of breastfeeding in this context [5, 6].

After the first year and a half of the pandemic, studies have reported that pregnant women or recently pregnant women with COVID-19, compared with non-pregnant women of reproductive age with COVID-19, are at greater risk of maternal death and more likely to be admitted to the intensive care unit or requiring invasive ventilation. Risk factors for severe COVID-19 in pregnancy are similar to those in other women of reproductive age, and include pre-existing comorbidities, non-white ethnicity, chronic hypertension, pre-existing diabetes, high maternal age, and high body mass index. Systematic reviews report that women with COVID-19 are found to be more likely to give birth to preterm babies [7]. However, studies are difficult to undertake given the different environments in which birth occurs and the variation in

participant selection, and risk status of the participants. Pregnant women remain a particularly vulnerable group at risk of mental health disorders during the pandemic due to the paucity of evidence on the possible effects of the virus on pregnancy. Added to this is the anxiety of not knowing about potential teratogenic effects of antivirals on the fetus [8].

The need to protect both midwives and the community from sites of contagion has led to hospitals being avoided [9, 10] and women and midwives reporting a rise in birth outside the conventional systems of care. However, reports have emerged from resource rich countries regarding a lack of capacity to provide home birth care [11]. Maintaining a physical distance in an attempt to reduce cross infection has led to radical changes in care during pregnancy, birth and the postnatal weeks. The midwife-woman relationship including meeting in person and providing a comforting touch is restricted [12]. Physical distancing and restrictions on travel may alleviate the stress on health-care systems but it may also have other unintended consequences for women and families including gender-based and family violence and psychological effects caused by isolation [13].

The pandemic has exposed the worlds' midwifery services to an added external level of disruption in addition to those that normally shape services such as the structures and resources of societies, communities, and health systems. In many countries, for example, Kenya, Tanzania and Uganda midwives play a major role in primary health care delivery, specifically in regions where other healthcare workers are scarce [9]. In some countries midwives have been redeployed away from providing their essential core services and women have been denied and actively discouraged from seeking care at health facilities. In Kenya, at the beginning of the outbreak media reports indicated that strict night-time curfews that confined women to their homes, resulted in the death of four women due to delays accessing emergency obstetric and newborn care [14]. In many other countries, the pandemic has added an extra threat to maternal survival amongst those women who may be already severely affected by economic welfare, societal inequalities and climate change [9, 15]. In addition to service challenges, midwifery education has also suffered challenges and setbacks in relation to social distancing and the closure of universities as well as the loss of clinical placement opportunities for students [16, 17].

The International Confederation of Midwives (ICM) represents 143 professional associations throughout the world. In early 2020 as the pandemic grew, ICM heard harrowing accounts from its member associations in all regions, and from the midwives working on the frontlines about increase in gender discrimination, domestic violence, human rights abuses, the over-medicalisation of birth, fear and misinformation, all culminating in growing distress among women and midwives. These findings were echoed by the United Nations Population Fund (UNFPA) and World Health Organisation (WHO) staff in country offices globally [14, 18, 19]. To explore this further, ICM coordinated this global research study to better understand the challenges and concerns of professional associations and colleges of midwives during the COVID-19 pandemic.

The recommendations from this research will provide valuable insights that may inform governments on future policy and 'best practice' to improve the safety and quality of maternity care following the pandemic and prepare the groundwork for dealing with future crises.

## Methods

The aim of the study was to gather information from midwifery associations across the world to determine the impact of the global pandemic. The study design is a descriptive cross-sectional online, web based, open-link survey. An email from the headquarters of ICM in The Hague was sent to every midwife association registered with ICM on the 1st of July 2020

inviting them to participate in the ICM COVID-19 survey. The participant information sheet, consent form and survey link to the survey was included. Participation was on a voluntary basis and a response to the survey implied consent. One response per association in either English, French or Spanish was invited. Online responses were automatically collected at a central data hub administered by ICM.

We asked the associations about the extent to which the midwifery profession was recognised by governments in planning a response to the pandemic or whether they felt they would be included in future planning. In addition, we invited the associations to outline how practices have changed and to what extent regulations, education and employment conditions have changed in response to the pandemic.

All responses were anonymous, and no attempt was made to verify the information put forward. The responses are the opinions of those who responded on behalf of their midwifery association. The study was performed and reported in line with the Checklist for Reporting Results of Internet E-Surveys (CHERRIES; www.jmir.org/2004/3/e34/). Ethical approval was received from Charles Darwin University in March 2020. HREC: H20105 –ICM COVID-19.

Follow-up in-depth interviews were conducted with those who nominated to be contacted in relation to the emerging themes. Case studies were called for from anyone wanting to participate on behalf of their country. Midwifery Associations could nominate that they were willing to have a follow up interview, however, this current paper only reports on the survey data.

## Setting

ICM has 143 Member Associations, representing 124 countries across every continent. Together these associations represent over 1 million midwives globally. ICM is organised into 6 regions divided into 10 sub regions (See Fig 1).

## Survey instrument

This survey was developed by the authors, a team of midwife researchers and clinicians and was critically reviewed in English. Prior to data collection, a pre-test of the survey was undertaken by a group of practicing midwives (n = 10) with the option for changes to be made to the draft questionnaire. The English version was translated by midwives who spoke French and

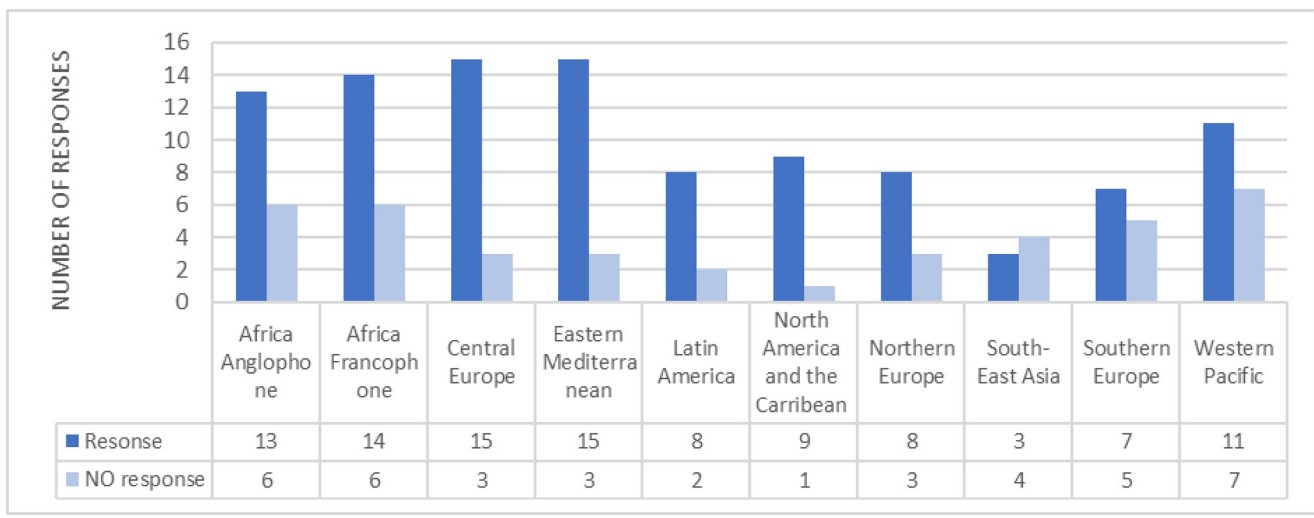

**Fig 1. Midwives' associations responses showing their geographical region.** (N = 143).

Spanish. The survey consisted of 106 questions divided into seven discreet sections and available in three languages. Each member association could select to answer in English, French or Spanish. The survey pertained to the time COVID-19 first arrived in each country and the subsequent months of living with the pandemic. The survey closed on the 1st April 2021. At the end of the survey, there was an opportunity, through free text, to identify the changes that have resulted since the onset of COVID-19.

The first section contained 32 questions relating to the issues specifically facing *midwives*. These questions related to what resources were available; were midwives informed appropriately of the dangers of the SARS-COV-2 virus; where did they access information; were there staff shortages and were midwives deployed outside maternity services? This section also sought information of the effect of COVID-19 on the midwives' families, their confidence to manage infected women and what happened to infected midwives?

The second section included 25 questions on the issues facing *women* that affected the response and practice of midwives. Questions were asked about the level of information and guidance available to women, whether they were afraid to go to maternity facilities to give birth, were they separated from planned support and their new-borns? Did women experience increased levels of surgical intervention such as induction and caesarean section? This section also included questions regarding the reduced access to maternity facilities that were closing to become COVID-19 facilities.

The third section contained 9 questions on the timeliness, accessibility, visibility and guidance from the *government departments or ministries of health*. Sections four and five questioned what *services* were affected (7 questions) and how *midwifery education* was affected (6 questions). The final sections, six and seven (20 questions) determined the associations responses to the pandemic in their country and whether there were benefits and changes that might eventuate in the future?

## Data collection, processing, and statistical analysis

Data were collected between July 2020 and April 2021 through an online survey platform (SurveyMonkey) and were available to the main authors of this article. All respondents fulfilling the inclusion criteria irrespective of whether they completed all questions in the survey were eligible for analysis. All data collected was anonymous. Data analysis was performed as an exploratory approach providing descriptive statistics (relative frequencies and percentages (n (%)). Multiple response questions were analysed as sum of answers per answer option (n (%)) and thus can exceed 100%. No inferential statistics were applied. Content analysis was used to explore responses to the open-ended questions and comments. All aspects of the survey were optional. Not all respondents answered every question, which means the amount of missing data varies.

## Identification of potential limitations of the study

No attempt was made to verify the responses given in the survey. Some member associations were not able to participate due to local acute health or political situations, local legislation, cultural or human rights restrictions or limited or disrupted access to information technology via the internet. All associations were assured of their anonymity in the reporting of the survey. In most cases the responses were made by a senior representative of the association, the president, or in many cases a nominated midwifery advisor of the college.

## Data collection methods

ICM sent an email invitation to their member Midwifery Associations to participate in the ICM COVID-19 survey with a participant information sheet and consent form available on an

online link. The email contained the link to the ICM COVID-19 survey of 106 questions. Confirmation that the midwifery association had read the participant information and consent forms was indicated via the initial question checkbox and the subsequent completion of the survey.

The data collection and analysis were independently conducted by authors who are independent of the ICM. This paper represents the responding ICM member Midwifery Associations views. The conclusions from the findings have been written in collaboration of other listed authors who participated in the conceptualisation of the study, development of the survey tool and the preparation of this manuscript.

## Results

The results of this survey are categorized into the seven different sections following the structure of the online questionnaire.

A total of 101 responses were received from the 143 surveys sent out to member organisations. Not all associations who responded answered all the questions. The number of responses received in each section is noted throughout. The geographical location of the associations who responded is shown in Fig 1.

### Section one–issues facing midwives

Many countries were caught unaware of the severity of the infection. Of the 87 associations who responded to this section of the survey, 23% of associations responded that their midwives had not been informed of the dangers at the outset of the pandemic and 68% said that they had been informed of the dangers of COVID-19 at the outset by government or Ministries of Health. In some countries a national strategy was in place and some midwives were offered personal protective equipment (PPE) at the outset of the pandemic. However, two thirds of associations responded that there was a lack of PPE for midwives and half said that they had to negotiate on behalf of their members for PPE to be made available to midwives (See Fig 2).

Although there had been a national strategy in place in many countries there was no consistent source of information available to associations. They therefore accessed information from a variety of sources including Ministries of Health, WHO, the media and ICM. Given the lack

| | YES N (%) | NO N (%) | NOT SURE N (%) | Responded N | Skipped N |
|---|---|---|---|---|---|
| **Was there a national strategy for providing PPE** | 62 (71%) | 17 (20%) | 8 (9%) | | |
| **Were midwives offered PPE at the outset** | 49 (56%) | 33 (38%) | 6 (5%) | | |
| **Was there a shortage of PPE for midwives** | 57 (66%) | 25 (29%) | 5 (6%) | | |
| **Did your Association negotiate for PPE for midwives** | 43 (49%) | 44 (51%) | | | |
| **TOTAL RESPONSES** | | | | 87 | 14 |

**Fig 2. The availability of PPE to midwives at the outset of the pandemic.** (N = 87).

of full PPE it is not surprising that 45% of the 87 associations who responded said their members did not feel confident that they were well protected in their workplaces.

The variation in the availability of PPE and other semi protective equipment was not confined to low- and middle-income countries (LMIC). The responses from associations regarding the availability of PPE and the perception that the midwives were well protected was as prevalent in low- and middle-income countries as it was in high income countries. When PPE was not available midwives resorted to all sorts of measures to try and keep themselves safe from the virus (See Fig 3).

From the text responses in the survey, we learnt that some midwives made protective equipment out of plastic garbage bags fashioned into gowns to protect themselves. Several associations said their midwives asked "patients or clients to go and buy some" protective clothing. Another association responded that the "midwives made videocalls instead of visits", another responded that midwives were proactive in making their own gear. Associations also reported that they sought donations from private sector organisations and helped each other out when there were donations.

Although 74% of associations reported that their members did receive training to manage infected women, less than half of those who responded said their members were advised how to provide virtual antenatal and postnatal care.

*"The training was sporadic and did not cover all the midwives across the country. It was through non-governmental organisation "* **#14**

It is not surprising to find that many midwives felt afraid to attend women in maternity facilities. Over half the associations (53%) said that midwives were afraid. The text responses also support this:

*"Fear was a factor, but generally there is strong accountability to show up for work, so midwives went to work"* **#5**

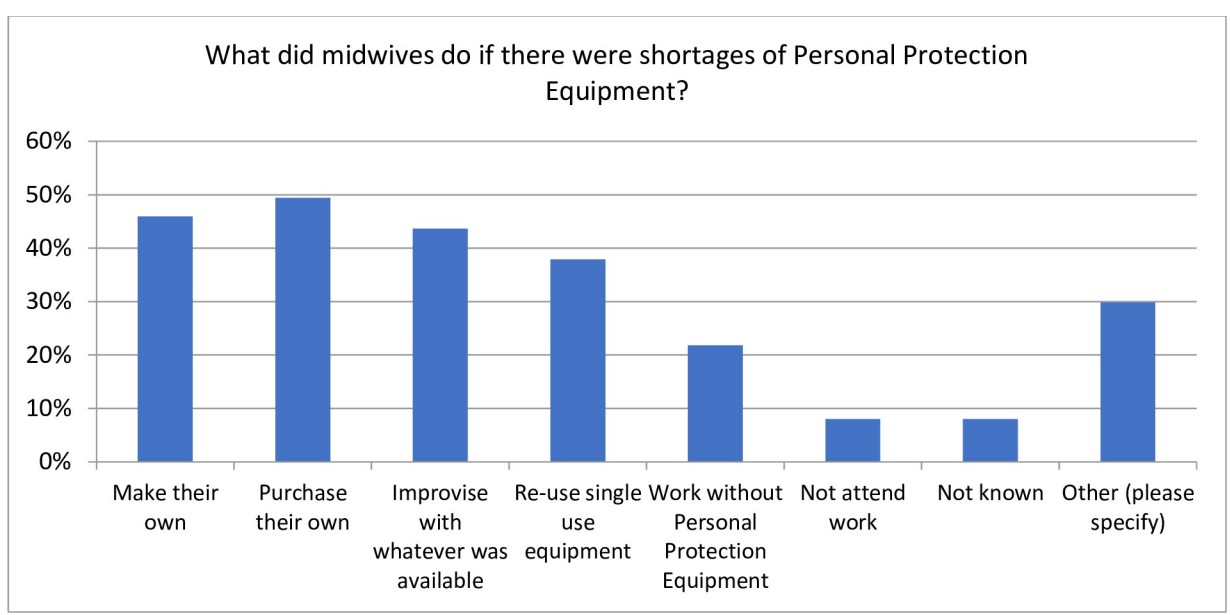

**Fig 3. The measures taken by midwives when there was NO PPE available.** (N = 87).

*"midwives were afraid to go to maternity facilities due to lack off PPE, there as well no policy which protects the rights of the midwives"* **#8**

*"Some Midwives expressed fear of going to work. For others it was just another duty, no fear expressed"* **#20**.

Across the world midwives have reported that during this pandemic they have been redirected into nursing roles to care for COVID-19 affected patients. Over half of the associations (53%) reported that the midwives had undertaken nursing duties, with (65%) of associations reporting that these duties were not the same as those undertaken before the onset of the pandemic. It is not known whether the deployment of midwives out of maternity facilities or the pre-pandemic shortages are reflected here, but over 60% of the associations who responded reported that there were midwife shortages. Again—it is not clear whether the midwife shortages reflect a pre-pandemic situation but having reported that there were shortages of midwives, (68%) also reported that the midwives on the ground worked much longer hours.

The pandemic continues to have a devastating effect on the lives of midwives around the world. At the time this survey was undertaken a third of the associations who responded could report that some midwives in their country had died. As the pandemic continues, this percentage will no doubt grow considerably.

## Section two–issues facing women

One of the most striking findings of this survey has been the change in the way women have had to respond to keeping themselves safe during the birth process. This is reflected in the following responses to whether women were afraid to go to maternity facilities, and whether there has been a reduction in attendance by women at maternity facilities.

Except for the questions on the availability of home birth (see below), 87 associations responded to this section of the survey. Over 75% reported that women were afraid to go to maternity facilities to give birth and a further 73% claimed this represented a serious reduction in attendance by women at maternity facilities during the pandemic.

In the words of one Spanish speaking respondent:

*"Sexual and reproductive health care has been made invisible . . .There was no security for women to attend the services and they also had to stay at home taking care of their children or the elderly, leaving their jobs."* #4

The associations who took part in this survey responded that there was a highly variable pattern regarding the availability of midwife care across the continuum of pregnancy labour and birth.

On the questions around home birth and community care 64 associations responded. Over 50% reported that there was an increased demand for community and home-based care from a midwife, and 40% responded that women were able to access home birth care. It appears that home birth and community-based services have filled a void during the pandemic as illustrated in the words of one association who responded:

*"Midwives were also asked to take on care of unattached moms and newborns while general practitioners offices were closed, to take on hospitalist shifts. . ."* #10

Although women received most of their information from various official sources including government websites and ministries of health, at least 70% accessed social media platforms for

all their information. Over half of the 87 associations who responded reported that there had been a lack of clear information and guidance for women from health ministries and the governments. Consequently, when the information source was unclear for women, over 85% accessed information from social media.

Breastfeeding advice was inconsistent at the beginning of the pandemic, however, as the pandemic continued, over 70% of associations reported there was no longer conflicting advice. Nevertheless, over half of the associations who responded said that women were concerned that they did not have information on how to protect their new-born babies.

Over a quarter of the respondents (26%) noted that some maternity facilities closed to become COVID-19 treatment facilities. It was also reported that many of these have not re-opened even though they are no longer required to remain open for COVID-19 patients. It was reported that some women were forced to give birth without a midwife due to a high percentage of facilities closing down to become COVID-19 facilities and midwives being at risk of becoming infected. Of those who responded, 18% said that they knew of occasions when women were not attended by a skilled birth attendant at birth and a further 14% were unsure.

*"As mentioned earlier, when the right to get midwifery care during a home birth were taken away, there might be some women who delivered without a midwife."* **#4**

*"(women) . . . were afraid of coming to hospitals."* **#6**

A great deal of media attention has been given to women's lack of choices and rights to sexual and reproductive health care during the pandemic. Among 87 associations, over half reported that women's rights have been denied. This is due in part to the major service change that has affected women's opportunity to give birth in the facility of their choice mainly due to the conversion of maternity units into COVID-19 facilities or a fear of becoming infected. Midwifery associations reported that women felt their rights were denied when they were not able to give birth in a place where they had originally planned to give birth. (This was supported by various mainstream media reports in many countries [21].) Overall women were directed to go to larger centralised maternity facilities. A further problem encountered during the pandemic has been the isolation of women from their partners and extended families. Sixty three percent (63%) of associations replied that women had been separated from having a support person at their birth. These sentiments were outlined in the words of one association:

*". . . maternity facilities restricted the number of support people to one per woman in labour. Many facilities required the support person to leave shortly after birth (within 2 hours) and did not allow women to have any visitors in the postnatal ward. . . (Culturally in this country) . . . birth is a social event and it is common for large family groups to support women in labour, wherever they give birth. Some women . . . felt that their right to have their partner or family member/s visit or stay with them was denied, and this led to a higher proportion of women discharging home early. Some maternity facilities allowed the support person to stay on with the woman in the postnatal ward, provided they stayed and did not come and go. This was well received by women and partners. We understand that the home birth rate increased due to women choosing to birth at home in order to have their chosen support people with them. While midwives mostly supported women's decisions to have a planned home birth, some midwives were concerned about infection prevention and control when family members from other household 'bubbles' expected to join the woman in her home to provide labour support"* **#22**

Forty percent (40%) of the 87 associations said women were denied access to contraception and 46% of associations reported that they expected there to be a rise in unplanned pregnancies. One of the most significant outcomes of the pandemic for women has been the global rise in domestic violence. Over 50% of the 87 associations reported a rise in domestic violence.

### Section three–the timeliness, accessibility, and visibility of information

Amongst the 84 associations who responded to this section, many aimed to keep their members informed from the start of the pandemic, however there were large inconsistencies in the way associations were provided and updated with information from the government or Ministries of Health. Concerningly, at least 19% of associations who responded received no advice at all concerning the pandemic (see Fig 4).

As the pandemic continued over 80% of the associations reported that information and updates were provided by government departments. In addition to this, associations reported that WHO and ICM provided further information to them, 80% and 70% respectively.

When further questioned regarding the quality and accessibility of the advice received 84 associations responded in the following way (see Fig 5).

### Section four–changes to services

In answer to the questions on service changes, of the 84 associations who responded to this section over half (55%) said that services had closed and another 54% said they had reduced their hours. These included sexual health services, family planning services, well child services and antenatal classes. In the words of one association

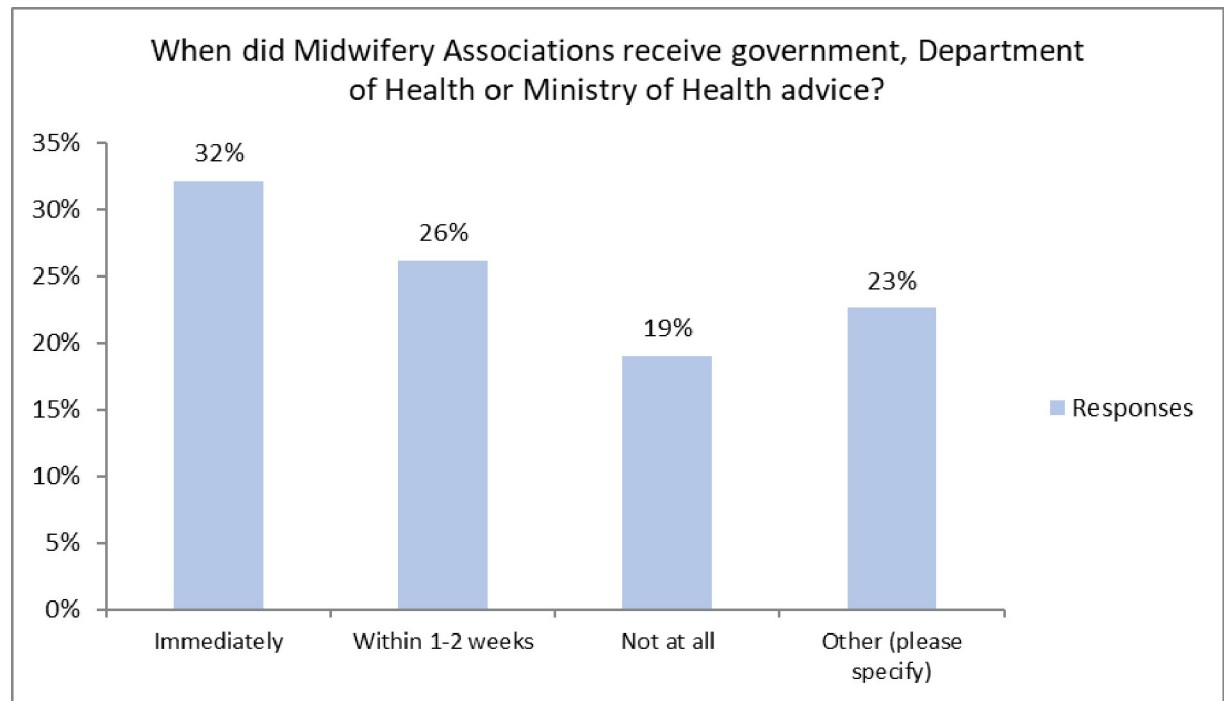

**Fig 4. The time between announcement of the pandemic and midwifery associations being notified and receiving advice from government.** (N = 84).

| Survey Question | YES<br>N (%) | NO<br>N (%) |
|---|---|---|
| Was there a Government/ Department of Health, Ministry of Health portal for practitioners to have questions answered? | 65 (75%) | 19(23%) |
| Was the guidance easy to find and accessible? | 76 (90%) | 8 (10%) |
| Was the guidance clear and explicit? | 71 (85%) | 13 (15%) |
| Did the Midwives' Association access information from the ICM website? | 73 (87%) | 11 (13%) |

**Fig 5. The quality and accessibility of advice to associations.** (N = 84).

*"lab services, hearing screening, ultrasound and mammography services, infant development programs, dentists (for frenectomy), public health parent/baby groups, lactation consultants were closed or decreased"* #13

When asked if midwives were required to offer care that was not usual in their country the associations responded in the following way:

*"Midwives took on additional community care . . . this included home visits, breastfeeding support, hearing screening. Midwives in some jurisdictions also took on COVID [sic] testing duties, lab services."* #5

There was also an increased demand for telehealth and e-consultation with 48% of associations responding that midwives were now required to offer telephone based antenatal and postnatal care and 46% responding that virtual or internet-based care had been introduced during the pandemic.

However, on the positive side:

*"the care a woman received in the COVID-19 ward was one on one midwifery care and a true blessing to come out of the pandemic"* #4

As well as the other duties noted in Fig 6, 84 associations who responded reported that midwives were also faced with "*a higher request for home births*", an increased demand for "*free standing birth centres*", as well as "*plenty of email requesting information about home birth services in the country*".

Associations also reported that midwives were having to discharge women earlier than usual (52%) and some (17%) with no follow up care. This trend towards earlier discharge appears to have also applied to women who gave birth via caesarean section, however there were fewer women who had no follow-up care (8%).

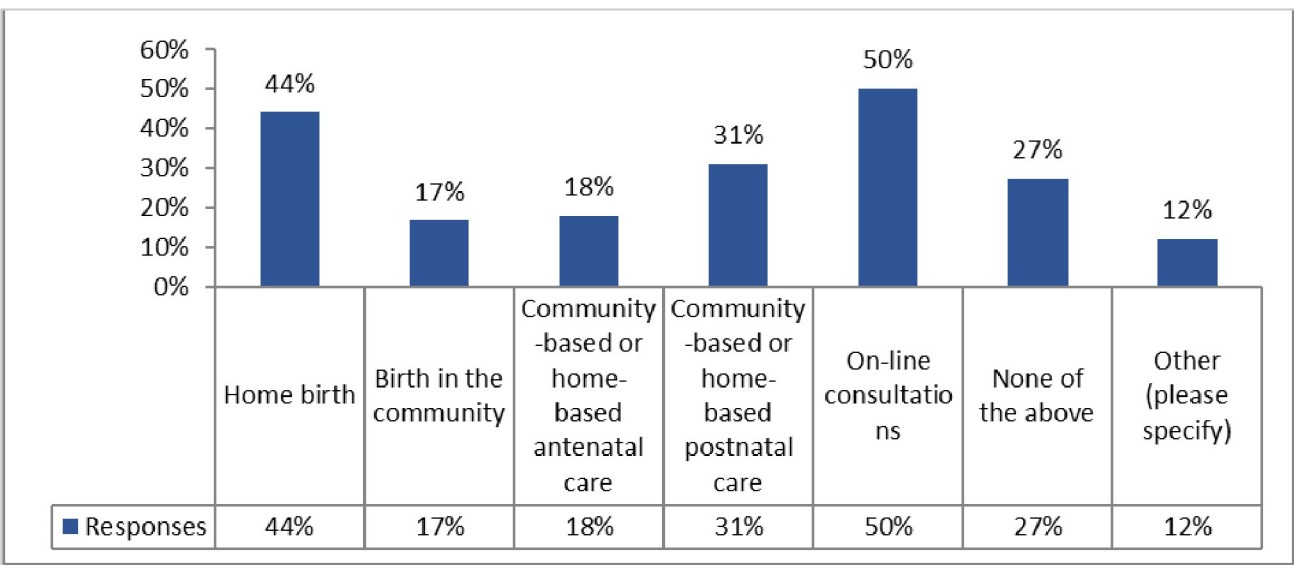

**Fig 6. The increased demand for midwifery services during the pandemic.** (N = 84).

### Section 5–changes to midwifery education

During the pandemic midwifery education has had to respond to the requirements to isolate and avoid personal contact to ensure the safety of midwifery students and the public. This has caused significant and overwhelming disruptions to midwifery education such as lack of clinical hours, lack of placements for students and increased use of simulation. Of the 83 associations who responded, 41% reported that midwifery education programmes had closed at the time of response. It is not known how many programmes have remained closed or how many have permanently replaced face to face teaching with online teaching. Responses regarding initial closures were from both high-income countries (HIC) and low to medium income countries (LMIC) countries. Such disruptions to midwifery education will have long term and very profound effects on the future numbers of registered and practising midwives worldwide.

In addition to programme closures, midwifery students are required to undertake alternative modes of learning with 93% of associations claiming that midwifery education has been moved to online learning. When questioned whether this may lead to significant delays in students being able to graduate and register as midwives over half of the 83 associations who responded (58%) agreed that this was the case. In the words of one association:

> *"There's been a strong push towards online lectures, tutorials and workshops. Decreased access to clinical placement and to continuity of care experiences with women. Depending on location might have been unable to attend any clinical placement"* #16

As well as the restriction of students into clinical areas, students have been expected to provide general nursing care. Nineteen percent (19%) reported that student midwives are also being asked to take on nursing roles for COVID-19 patients, even though they do not always have the training or expertise to provide this care.

Other challenges that have been exacerbated by the closure of midwifery education courses are noted here:

*"Funding challenges, because classes were suspended students couldn't access student loans. Housing challenges because students had no employment and no student loans. Despite these challenges many students organised and volunteered en mass to support registered midwives during the pandemic with collating resources or sewing and delivering PPE"* #5

### Sections six and seven: What does the future hold?

When asked to comment on the future of midwifery in their countries, 79 associations responded. Midwives' associations have played a major role in advocating for the safety of their members, with 84% reporting that they had advocated to authorities for the provision of PPE for midwives and a further 58% advocated for COVID-19 testing for their members.

Associations were asked for the specific issues they were involved in on behalf of their members during the pandemic (see Fig 7A and 7B).

In addition to these requests several associations responded:

*"none of the above"* #70

*and*

*"the situation was so confused and like war situation that all the professionals were trying to do the best for the clients."* #62

When asked whether associations had been consulted or invited to contribute to government policies relating to COVID-19 only a little over half of those who responded said they had been invited to contribute sometimes (48%) or all of the time (16%). Perhaps not surprisingly 35% had not been consulted at all.

The response of governments in inviting associations to contribute to national health planning was similarly disappointing with under 50% of those who responded being invited to participate. However, associations clearly made an effort to be invited to the policy table.

*"Although we were not invited, we participated and insisted on quality care and protection of women and midwives. We organised meetings with institutions."* **#2**

*"we invited ourselves"* **#5**

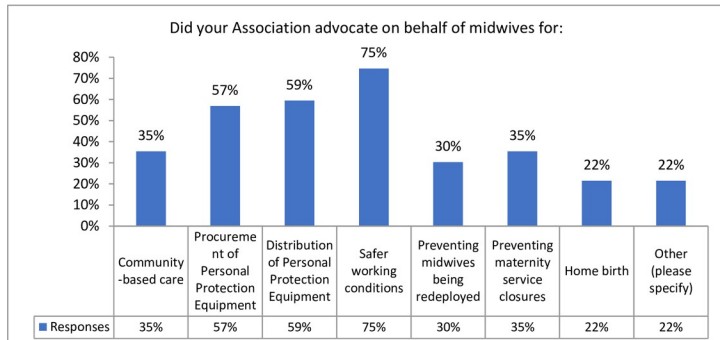
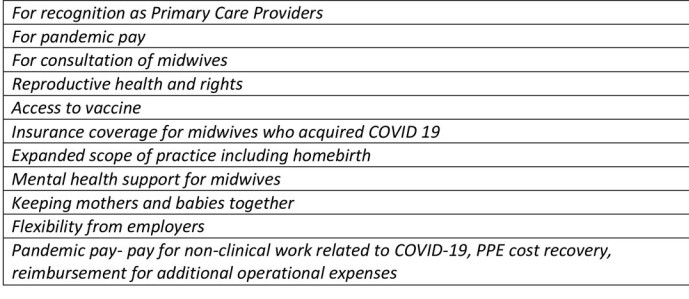

**Fig 7.** a. The specific issues advocated for on behalf of midwives (N = 79). b. Summary of the Text responses for specific issues advocated for on behalf of midwives (N = 79).

Where associations were consulted, it was with the Health Departments or Ministry of Health (55%) and a further 18% with government. However, the associations reported that in over a third of cases (37%) they were not consulted at all.

In terms of being invited to contribute to national health policy through representation on national health planning committees, almost half of the associations who responded said they had not been invited to participate. However, associations clearly made an effort to be invited to the policy table and 52% of those who responded did call for legislative changes, such as a call to be included in pandemic planning, a call to provide more community and homebased care and for funding for telehealth (see Fig 8).

On the subject of how midwives feel about the future, equal number of associations painted positive and gloomy predictions for the future:

> _Positive:_ "More effective use of technology and virtual meetings / consultations and networking with colleagues.

> _Gloomy:_ "Greater fear of accessing and using healthcare facilities at present. Midwives and health care workers are exhausted, more so than at the beginning Increasing amounts of concern around mental health and wellbeing for all healthcare workers. concerns about numbers of midwives who may choose to leave the profession or retire because of burnout, exhaustion and Post traumatic distress disorder." **#50**

The window on the world of midwives and childbearing women through the eyes of the professional associations representing midwives paints a powerful and worrying picture. Midwives form a predominantly female workforce, often juggling competing responsibilities of children and family care who nevertheless claim that their professional Associations 'are needed now more than ever before'. This closing comment sums up the situation:

> "Midwives levels of burnout have increased. Their mental health has decreased. Many are facing financial stressors.

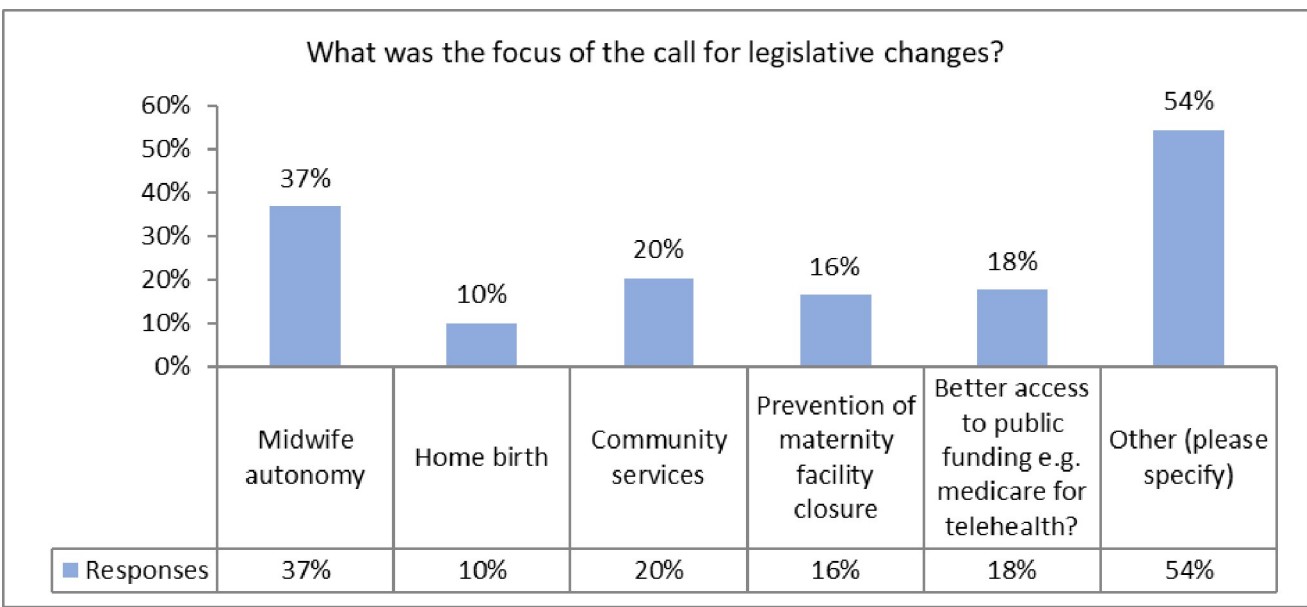

**Fig 8. The focus for legislative changes called for by midwives' associations during the pandemic.** (N = 79).

*Midwives have adopted telehealth/virtual health.*

*Midwives now offer an altered prenatal and postnatal visit schedule with fewer visits. . .*

*Women giving birth today conceived during COVID [sic] they have never seen our faces; we have never seen theirs. Often clients attend visits alone, unsupported by their partner(s), without including their children in the magic of their care. Perinatal anxiety and depression anecdotally appear to be on the rise.*

*Midwives wear PPE for all client encounters, those most cautious changing their scrubs in their cars, parking lots, or garages so they come home in clean clothes, they shower right away before they hug their children. . .*

*Midwives are working more: meetings, policy development, infection prevention and control, tech set up, workforce and human resources planning and increased communication with clients amounts to at least 10 additional unpaid hours of work per midwife per month. . .*

*Midwives report operational costs are up. . . many midwives have experienced increased childcare costs with the closure of schools and day-care.*

*Some midwives have had to step away from clinical practice due to their health (immunocompromised) or to protect the health of an immunocompromised family member.*

*Many midwives have had to take unpaid leave and sometimes have been out of pocket to hire a locum to cover them while they get COVID [sic]testing and await results, self-isolate due to exposure or recover from COVID [sic]. Other midwives are working extra to cover for their colleagues who are on leave.*

*. . . those who acquire COVID [sic] and suffer long term sequela may not be covered by short- or long-term disability or may not receive the benefit of the doubt from their insurers that it was a work acquired illness.*

*Out of hospital birth and community care has become more sought after than ever. Midwives reduce pressure on hospital services, help to prevent infection by keeping women out of acute care facilities, and have stepped up to fill gaps in care when physicians and nurses closed their doors or were redeployed.*

*Midwives, especially in rural areas, have had trouble accessing registration renewal.*

*Midwives perceived their burnout as being related to their work, increased demands during COVID [sic[, and again a lack of recognition and support from the government. . ." #54*

## Discussion

This study is a collection of responses from the professional organisations that represent midwives across the world under the umbrella of the International Confederation of Midwives. Together these responses paint a picture of the impact of the global pandemic on maternity care through the eyes of midwives. The study identified the sheer scale of many of the global issues facing midwives [20] and women from the start of the pandemic until midway through 2021. The common themes included a lack of supply of PPE, the move to online and telephone consultations in addition to the uncertainty of where to give birth in the context of fear of infection and the changing status of maternity facilities becoming COVID-19 facilities. Associations also reported a reduction in face-to-face consultations in antenatal and postnatal care and a strong emphasis on the changing role of community and home birth care. Many

associations reported the loss of women's autonomy in choosing their place of birth and of having a chosen support person with them during birth and postnatally. This was supported by various mainstream media reports in many countries [21]. There appears to be an alarming increase in violence to women. These factors translate to a denial of human rights for child-bearing women in many instances. The reality of burnout, exhaustion and declining mental health in the health workforce has been identified across the world in relation to COVID-19 [22]. Our survey has shown that all of these issues are impacting the midwifery workforce, and exist across all regions, globally. They are widespread and are not limited to high- or low-income countries. In many countries midwives have been working without pay and in risk high-risk situations that threaten their own life as well as those of colleagues and family members. In addition to this the education of midwives has been severely disrupted, in some places closing down altogether. This will surely exacerbate the already existing midwife shortages worldwide.

Researchers publishing over the past eighteen months [12] echo many of the issues raised by midwifery associations in this survey. Other researchers point to the dangers of reverting to old patterns of command and control [23]. Solutions include the recognised safety of midwife-led units and birth in community settings [14, 24] and a measured strategic response to future epidemics [18].

## Recommendations

Recommendations from this current survey parallel those of the Global Call to Action by ICM and UNFPA [19] and the WHO Independent Panel for Pandemic Preparedness and Response report prepared for the World Health Assembly in May 2021 [18]. Amongst their recommendations the WHO Independent panel [18] concluded that midwives must be visible to ministries of health and governments and that midwife-led continuity of care must be prioritised, with greater support provided for community-based midwifery care. Lack of access to sexual and reproductive health services has long-term, wide-ranging negative implications for individuals and society and midwives play a central role in upholding and protecting women's rights. The Global call to Action [19] supported these recommendations as well as calling for the equitable availability of PPE for midwives, ensuring midwives and birthing centres are properly equipped to deliver quality care and prioritise testing for COVID-19 for all pregnant women, as well as the midwives who care for them.

Policy makers should ensure midwife involvement and leadership in determining health policy and effective pandemic responses, recognising that midwives are the most appropriate professionals to inform the government about effective organisation of midwifery services, and of their own needs and those of the women and newborns they care for.

Midwifery care and nursing care are not interchangeable, therefore midwives should not be deployed to areas outside of their scope of practice, unless imperative. Neither should child-bearing women be left without a qualified midwifery workforce to provide respectful, competent and safe maternity care. If midwives are going to be able to offer women accurate information regarding COVID-19 they must have access to evidence-based guidance, training and other COVID-19 resources.

The closure of maternity services, and not allowing a birth companion even where infection prevention and control measures are in place, separating mother and newborn after birth, not permitting breastfeeding or contact between mother and newborn, and enforced medical interventions such as unnecessary caesarean or induction of labour are all blatant violations of women and newborn's rights during pregnancy and birth. They are also counter to the evidence on safe and effective care.

Governments and all those in authority are fully informed of the increased risk of sexual and gender-based violence, particularly domestic violence that women and midwives face during a crisis. An essential part of crisis and emergency interventions should include recognition of the importance of safe maternity care alongside the provision of family planning and safe post-abortion.

This survey sheds some light on the lack of basic resources and the inadequacy of current health systems to deal with a global pandemic. Midwives feel overworked and underpaid in health systems where there are inadequate training opportunities, and restrictive policies. For the midwifery profession, the chronic problems currently manifesting run especially deep. The midwifery profession has struggled to gain access to funding, resources, training, and recognition as an autonomous profession well before the onset of COVID-19.

Compounding this situation is the rising rate of women seeking care from midwives outside of facilities, including in countries where community-based midwifery services are not part of usual maternal and newborn services. Women are fearful of birthing in hospitals where they risk infection. Women are being discharged within hours of giving birth, including women who give birth by caesarean section. Often no follow-up care is available. Midwives are stepping up to provide some of this care to women and their newborn, but they are often not resourced for this work.

Solutions lie in removing structural barriers to enable funding streams, financial drivers and insurance mechanisms to directly allocate funds to maintaining midwifery services. This includes enabling community-based services that are crucial at a time when facility-based services are decreasing and when women are increasingly opting for decentralised services. Disseminating funds directly to the organisations that represent midwives as frontline maternal healthcare providers is the best way to ensure they have the resources and capacity to provide community-based services and enable midwife-led care to reach the most vulnerable women in communities.

The WHO report warns future catastrophes can only be avoided by a change in preparedness, a commitment to new systems that are co-ordinated, connected and accountable [18]. Other researchers [24] call for phased strategic planning including both national and local responses to testing and treatment capacity. Midwifery associations and women are calling for safer maternity care which includes care in the community and in midwifery units where exposure to sick and symptomatic patients is minimized [19].

## Conclusions

Our survey is a snapshot of the many problems faced by midwives in a predominantly female workforce during the COVID-19 pandemic. Many of these problems are currently being more closely examined and researched. Two major factors that this survey identified that must be urgently addressed however, are the lack of representation of the midwifery profession in drawing up government policy on the strategic responses to new epidemic threats in maternity care; and the continued denial of the conclusive evidence for the safety of out of hospital birth, either in homes or in the community in midwife led birth centres. Both these recommendations stand to enhance the effectiveness of midwives in a world that is currently being ravaged by the SARS-COV-2 virus and a future world that may face similar catastrophic pandemics.

## Supporting information

**S1 File. ICM COVID 19 survey (English).**
(PDF)

## Acknowledgments

The authors would like to acknowledge the midwives of the world and the ICM Midwifery Association Members for their amazing work and endurance during the Global COVID 19 pandemic and for continuing to care for mothers, babies, families and fellow midwives in some of the most trying times and conditions.

## Author Contributions

**Conceptualization:** Donna L. Hartz, Sally K. Tracy, Sally Pairman, Ann Yates, Charlotte Renard, Pat Brodie, Sue Kildea.

**Data curation:** Donna L. Hartz, Sally K. Tracy.

**Formal analysis:** Donna L. Hartz, Sally K. Tracy.

**Funding acquisition:** Sally Pairman.

**Investigation:** Donna L. Hartz, Sally K. Tracy.

**Methodology:** Donna L. Hartz, Sally K. Tracy, Sally Pairman, Pat Brodie, Sue Kildea.

**Project administration:** Donna L. Hartz, Sally K. Tracy, Sally Pairman, Charlotte Renard.

**Resources:** Sally Pairman.

**Supervision:** Sally K. Tracy, Sally Pairman, Sue Kildea.

**Validation:** Donna L. Hartz, Sally K. Tracy, Sally Pairman, Ann Yates, Charlotte Renard, Pat Brodie, Sue Kildea.

**Writing – original draft:** Donna L. Hartz, Sally K. Tracy, Sally Pairman, Ann Yates, Charlotte Renard, Pat Brodie, Sue Kildea.

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
