## [Decision Letter · Decision Letter 0]

29 Jul 2022

PONE-D-22-06656Midwives Speaking Out on COVID-19: International Confederation of Midwives (ICM) Global SurveyPLOS ONE

Dear Dr. Hartz,

Thank you for submitting your manuscript to PLOS ONE. After careful consideration, we feel that it has merit but does not fully meet PLOS ONE’s publication criteria as it currently stands. Therefore, we invite you to submit a revised version of the manuscript that addresses the points raised during the review process.

Please note that we have only been able to secure a single reviewer to assess your manuscript. We are issuing a decision on your manuscript at this point to prevent further delays in the evaluation of your manuscript. Please be aware that the editor who handles your revised manuscript might find it necessary to invite additional reviewers to assess this work once the revised manuscript is submitted. However, we will aim to proceed on the basis of this single review if possible.

The reviewer has raised a number of minor editorial concerns that need attention. Could you please revise the manuscript to carefully address the concerns raised?

We look forward to receiving your revised manuscript.

Kind regards,

Sebastian Shepherd

Staff Editor

PLOS ONE

Journal Requirements:

We also acknowledge the financial support of UNFPA and Johnson and Johnson™.

Contracted project.

United National Population Fund (UNFPA): https://www.unfpa.org

UNFPA played no direct role in the undertaking of this research. A UNFPA representative (Dr Bar-Zeev) was member of the steering committee and due to her unique role in maternal health in UNFPA contributed to the study design, survey tool development and manuscript authorship.

Reviewers' comments:

Reviewer's Responses to Questions

**Comments to the Author**

1. Is the manuscript technically sound, and do the data support the conclusions?

Reviewer #1: Yes

2. Has the statistical analysis been performed appropriately and rigorously? 

Reviewer #1: Yes

3. Have the authors made all data underlying the findings in their manuscript fully available?

Reviewer #1: No

4. Is the manuscript presented in an intelligible fashion and written in standard English?

Reviewer #1: Yes

5. Review Comments to the Author

Reviewer #1: Thank you for the opportunity to review this interesting paper. Below are minor editorial suggestions but overall the manuscript is clear and well written. The research methodology sound.

Abstract:

L54 Consider changing do to did – for the context of the results of this study “More than 50% of the associations reported that governments do not consult them,”

Background

L64 and thereafter WHO is spelt with a Z in Organization

L81 “at greater risk of maternal death and more likely to” than what/who? Do you mean pregnant women with covid compared to non-infected or pregnant women with covid than other non-pregnant people with covid?

L95 and abstract you have used homebirth and home birth – please choose one and be consistent

L112-117 avoid single sentence paragraphs

L129-133 is method not background

L148 survey responses may have been anonymous but not interviewees

L167 “Midwifery Associations could nominate that they were willing to have a follow up interview,” associations are not human – the associations representative or respondent could nominate that they were willing to have a follow up interview,

L171 and elsewhere you use both SARS COV 2 and SARS-CoV-2 - – please choose one and be consistent, similarly COVID 19 and COVID-19 and COVID

L181-186 avoid single sentence paragraphs

Suggest you preface the sections part in saying there were seven sections to the survey. And then put all in one paragraph

L188 tool or platform?

L194 “Quantitative data were analysed using descriptive statistics” is repetitive and I don’t think needed

L209 “Survey Monkey™ platform. C” repetitive as already stated

L213 “United Nations Population Fund (UNFPA).” Already qualified abbreviation so just use UNFPA here

L215 sentence starting “This paper and represents their views and conclusions from the findings” does not make sense

L225 Table 1. Midwives’ association responses showing their geographical region (N=101) – this actually includes responders and non-responders so n should be 143

L292 italicise first word

L299 remove brackets around 40%

L319 remove brackets around 18%

L320 remove brackets around 14%

L330-331 “This was supported by various mainstream media reports in many countries [21]” is discussion not survey finding

L363 consider changing wording or ‘wore on’ to something like continued

L390 add quote mark after births

L405 I don’t think HIC and LMIC have been qualified

L445 remove brackets around 35%

L454 remove brackets around 18%

L459 remove brackets around 52%

L512 capital I in international

L548 suggest changing COVID-19 to pandemic – to capture future events

6. PLOS authors have the option to publish the peer review history of their article (what does this mean?). If published, this will include your full peer review and any attached files.

Reviewer #1: **Yes: **Linda Sweet

---

## [Author Response · Author response to Decision Letter 0]

27 Sep 2022

Reviewer Responses- All of this information is inlcuded below and in the table in the document "response to reviewers".

Reviewer #1: Thank you for the opportunity to review this interesting paper. Below are minor editorial suggestions but overall, the manuscript is clear and well written. The research methodology sound. 

Answer: Thank you for your time, Professor Sweet in reviewing this paper and your considered, thorough approach, as always.

Abstract:

L54 Consider changing do to did – for the context of the results of this study “More than 50% of the associations reported that governments do not consult them,” 

Answer: L55 ‘do’ has been replaced with ‘did’

Background 

L64 and thereafter WHO is spelt with a Z in Organization Answer: Organization replaced with organisation

All other uses of the letter z in organise, organised and organisation have been replaced.

L81 “at greater risk of maternal death and more likely to” than what/who? Do you mean pregnant women with covid compared to non-infected or pregnant women with covid than other non-pregnant people with covid?

Answer: L82-83 This is clarified with the following statement

…pregnant women or recently pregnant women with COVID-19, compared with non-pregnant women of reproductive age with COVID-19…

L95 and abstract you have used homebirth and home birth – please choose one and be consistent

Answer: The authors have chosen home birth and this term is now consistent throughout the paper.

L112-117 avoid single sentence paragraphs

Answer: These sentences have now been incorporated into the previous paragraph. L 113-115.

L129-133 is method not background Answer: The sentence has now been moved to the Methods section. L144-145

L148 survey responses may have been anonymous but not interviewees 

L167 “Midwifery Associations could nominate that they were willing to have a follow up interview,” associations are not human – the associations representative or respondent could nominate that they were willing to have a follow up interview, 

Answer: Thank you for highlighting this inconsistency. L154-158 - These sentences have been moved to a new stand-alone paragraph as this relates to methods and data collection and it is stated that the interview data are not analysed or reported on in this paper.

L171 and elsewhere you use both SARS COV 2 and SARS-CoV-2 - – please choose one and be consistent, similarly COVID 19 and COVID-19 and COVID

Answer: The authors have edited the inconsistencies and have used SARS-COV-2 and COVID-19.

Where COVID has been used in a quotation this has been followed by [sic].

L181-186 avoid single sentence paragraphs

Suggest you preface the sections part in saying there were seven sections to the survey. And then put all in one paragraph

Answer: L186-190 The sentences have been amalgamated into one paragraph.

L188 tool or platform? Answer L192 Tool has been replaced with platform 

L194 “Quantitative data were analysed using descriptive statistics” is repetitive and I don’t think needed. Answwer This has been removed

L209 “Survey Monkey™ platform. C” repetitive as already stated. Answer: This has been removed.

L213 “United Nations Population Fund (UNFPA).” Already qualified abbreviation so just use UNFPA here. Answer: At the request of the editors this has been removed from this section.

L215 sentence starting “This paper and represents their views and conclusions from the findings” does not make sense

Answer: L215-217

Answer: Replaced with: 

This paper represents the responding ICM member Midwifery Associations views. The conclusions from the findings have been written in collaboration of the authors who participated in the conceptualisation of the study, development of the survey tool and the preparation of this manuscript.

L225 Table 1. Midwives’ association responses showing their geographical region (N=101) – this actually includes responders and non-responders so n should be 143. Answer: L 226 Changed to (N=143)

L292 italicise first word. Answer: L293 Word now italicised

L299 remove brackets around 40%

L319 remove brackets around 18%

L320 remove brackets around 14% Answer: Brackets removed L300, L320 and L321.

L330-331 “This was supported by various mainstream media reports in many countries [21]” is discussion not survey finding. Answer: Moved to discussion section L524-525.

L363 consider changing wording or ‘wore on’ to something like continued. Answer: L312, 364 wore on changed to continued

L390 add quote mark after births Answer: L391 quote marks inserted after births

L405 I don’t think HIC and LMIC have been qualified. Answer: L406-407 These acronyms have now been qualified: high-income countries (HIC) and low to medium income countries (LMIC) countries

L445 remove brackets around 35%

L454 remove brackets around 18%

L459 remove brackets around 52% Answer: Brackets removed L447, L456, L461.

L512 capital I in international Answer: L515 ‘I’ now capitalised

L548 suggest changing COVID-19 to pandemic – to capture future events. Answer L552: COVID-19 changed to pandemic.

---

## [Decision Letter · Decision Letter 1]

7 Oct 2022

Midwives Speaking Out on COVID-19: The International Confederation of Midwives Global Survey

PONE-D-22-06656R1

Dear Dr. Hartz,

We’re pleased to inform you that your manuscript has been judged scientifically suitable for publication and will be formally accepted for publication once it meets all outstanding technical requirements.

Kind regards,

Forough Mortazavi

Academic Editor

PLOS ONE

Additional Editor Comments (optional):

Reviewers' comments:

Reviewer's Responses to Questions

**Comments to the Author**

1. If the authors have adequately addressed your comments raised in a previous round of review and you feel that this manuscript is now acceptable for publication, you may indicate that here to bypass the “Comments to the Author” section, enter your conflict of interest statement in the “Confidential to Editor” section, and submit your "Accept" recommendation.

Reviewer #1: All comments have been addressed

2. Is the manuscript technically sound, and do the data support the conclusions?

Reviewer #1: Yes

3. Has the statistical analysis been performed appropriately and rigorously? 

Reviewer #1: Yes

4. Have the authors made all data underlying the findings in their manuscript fully available?

Reviewer #1: No

5. Is the manuscript presented in an intelligible fashion and written in standard English?

Reviewer #1: Yes

6. Review Comments to the Author

Reviewer #1: Thank you for addressing the minor edits raised. I think this is important work and will be of value to the midwifery community

7. PLOS authors have the option to publish the peer review history of their article (what does this mean?). If published, this will include your full peer review and any attached files.

Reviewer #1: **Yes: **LINDA SWEET

---

## [Editor Report · Acceptance letter]

20 Oct 2022

PONE-D-22-06656R1 

Midwives Speaking Out on COVID-19: The International Confederation of Midwives Global Survey 

Dear Dr. Hartz:

I'm pleased to inform you that your manuscript has been deemed suitable for publication in PLOS ONE. Congratulations! Your manuscript is now with our production department. 

Kind regards, 

on behalf of

Dr. Forough Mortazavi 

Academic Editor

PLOS ONE